# Effective Diagnosis of Foot-And-Mouth Disease Virus (FMDV) Serotypes O and A Based on Optical and Electrochemical Dual-Modal Detection

**DOI:** 10.3390/biom11060841

**Published:** 2021-06-05

**Authors:** Yun-Jung Hwang, Kyung-Kwan Lee, Jong-Won Kim, Kwang-Hyo Chung, Sang-Jick Kim, Wan-Soo Yun, Chang-Soo Lee

**Affiliations:** 1Bionanotechnology Research Center, Korea Research Institute of Bioscience and Biotechnology (KRIBB), Daejeon 34141, Korea; yunjung1758@gmail.com (Y.-J.H.); lkk@kribb.re.kr (K.-K.L.); kimjw@kribb.re.kr (J.-W.K.); 2Department of Chemistry, Sungkyunkwan University, Suwon 16419, Korea; 3Department of Biomedical and Nanopharmaceutical Science, College of Pharmacy, Kyung Hee University, Seoul 02447, Korea; 4Dignostics Platform Research Section, Electronics and Telecommunications Research Institute (ETRI), Daejeon 34129, Korea; hyo@etri.re.kr; 5Synthetic Biology and Bioengineering Research Center, Korea Research Institute of Bioscience and Biotechnology (KRIBB), Daejeon 34141, Korea; sjick@kribb.re.kr; 6Department of Biotechnology, University of Science & Technology (UST), Daejeon 34113, Korea

**Keywords:** dual-modality, optical, electrochemical, foot-and-mouth disease virus (FMDV)

## Abstract

Foot-and-mouth disease virus (FMDV) is a highly contagious disease that affects cloven-hoofed animals. The traditional diagnostic methods for FMDV have several drawbacks such as cross-reactivity, low sensitivity, and low selectivity. To overcome these drawbacks, we present an optical and electrochemical dual-modal approach for the specific detection of FMDV serotypes O and A by utilizing a magnetic nanoparticle labeling technique with resorufin β-d-glucopyranoside (res-β-glc) and β-glucosidase (β-glc), without the use of typical lateral flow assay or polymerase chain reaction. FMDV serotypes O and A were reacted with pan-FMDV antibodies that recognize all seven FMDV serotypes (O, A, C, Asia 1, SAT 1, SAT 2, and SAT 3). The antigen–antibody complex was then immobilized on magnetic nanoparticles and reacted with β-glc-conjugated FMDV type O or type A antibodies. Subsequently, the addition of res-β-glc resulted in the release of fluorescent resorufin and glucose owing to catalytic hydrolysis by β-glc. The detection limit of fluorescent signals using a fluorescence spectrophotometer was estimated to be log(6.7) and log(5.9) copies/mL for FMDV type O and A, respectively, while that of electrochemical signals using a glucometer was estimated to be log(6.9) and log(6.1) copies/mL for FMDV type O and A, respectively. Compared with a commercially available lateral flow assay diagnostic kit for immunochromatographic detection of FMDV type O and A, this dual-modal detection platform offers approximately four-fold greater sensitivity. This highly sensitive and accurate dual-modal detection method can be used for effective disease diagnosis and treatment, and will find application in the early-stage diagnosis of viral diseases and next-generation diagnostic platforms.

## 1. Introduction

Foot-and-mouth disease (FMD) is a highly transmissible and fatal disease of wild and domestic cloven-hoofed animals such as cattle, sheep, goat, and swine. It is caused by Foot-and-mouth disease virus (FMDV) (genus *Aphthovirus*, family *Picornaviridae*) and has high morbidity and low mortality rates in infected animals. As FMDV can disseminate over long distances and cause acute epidemics in FMD-free areas, outbreaks of FMD severely restrict international trade in animals and related materials, triggering massive economic damage [1]. Therefore, it is necessary to diagnose FMD quickly and efficiently in the field. FMDV is a small, non-enveloped, and positive-sense RNA virus [2]. It has seven immunologically distinct serotypes, namely, O, A, C, Asia 1, Southern African Territories (SAT) 1, SAT 2, and SAT 3, with a diverse antigenic spectrum of strains within each serotype [3]. FMDV types O and A are the most prevalent worldwide and have spread widely in South Korea since the early 2000s [4]. Thus, the early diagnosis of FMDV types O and A is of particular importance.

Various in vitro diagnostic methods have been developed for FMDV detection, including virus isolation, antigen enzyme-linked immunosorbent assay (Ag-ELISA) [5], lateral flow assay (LFA) [6], reverse transcription–polymerase chain reaction (RT-PCR) [7,8,9,10,11,12,13,14], and reverse transcription–loop-mediated isothermal amplification (RT-LAMP) [15,16,17]. Recently, several studies have focused on molecular diagnostic methods to detect viral nucleic acids based on RT-PCR and RT-LAMP. PCR is the most powerful method owing to its high sensitivity through gene amplification of the target DNA. However, PCR tests have limited efficiency as they require time-consuming and temperature-dependent denaturation, annealing, and elongation steps. Moreover, PCR tests frequently generate false-positive results [18,19].

LFAs are a well-established and valuable tool for point-of-care testing in the biomedicine, agriculture, food, and environmental sciences fields, as they are inexpensive, easy to use, and portable [19]. Moreover, they provide rapid results. Nevertheless, LFAs have a complex structure, which means that several components must be considered when designing the strips. Furthermore, LFAs only provide qualitative (on/off) or semi-quantitative results, which means they are only suitable for primary screening. Likewise, traditional FMDV detection using LFAs has serious drawbacks with regard to sensitivity, specificity, and cross-reactivity. Highly sensitive, specific, and rapid virus detection is a cornerstone for the accurate diagnosis and control of a variety of infectious viruses, including FMDV [20]. Therefore, recent advances in fundamental features of LFAs have included new signal amplification strategies, nanoparticle labeling, quantification systems, and methods for the simultaneous detection of multiple serotypes [21,22].

Recently, various approaches have emerged for efficient virus detection based on the signal outputs of different chemical and biological sensors. These methods, including surface-enhanced Raman spectroscopy (SERS), fluorescence, electrochemistry, and colorimetry [23], have received considerable attention for early diagnosis and real-time monitoring. Although these technologies each have certain advantages, no single technique can provide enough information for an efficient diagnosis due to inherent shortcomings in sensitivity, multiplexing capabilities, and response times [24,25]. On the contrary, dual- or multi-modal sensor platforms that measure two or more output signals by using one or more probes have an inherent advantage over conventional single signal amplification platforms, in that they can ensure enhanced diagnostic accuracy by data coupling, mutual verification, and the elimination of interference. For example, the detection of both fluorometric and colorimetric signals facilitated the highly sensitive and multifunctional detection of aptamer, arginine, and thrombin in a complex matrix [26]. Several studies have demonstrated the detection ability, reliability, sensitivity, and selectivity of dual- or multi-modal sensors, which highlights their potential for use in real analyses [27,28,29,30,31,32,33].

Herein, we report a dual-modal sensing platform for FMDV detection via an immunoassay using resorufin-β-d-glucopyranoside (res-β-glc) and β-glucosidase (β-glc) that produces both fluorescent and electrochemical signals. Res-β-glc is a stable fluorogenic galactosidase substrate that generates fluorescent resorufin and glucose molecules upon interaction with enzymes such as β-glc. β-glc catalytically hydrolyzes the glycosidic bonds in res-β-glc to form a terminal non-reducing residue of β-d-glucosides and oligosaccharides, followed by the release of resorufin and glucose molecules, which can be detected by optical and electrical analyses, respectively. Magnetic nanoparticles (MNPs) conjugated with pan-FMDV antibodies (pan-Ab), which can react with all FMDV serotypes, were reacted with different concentrations of FMDV type O or A. Then, β-glc-conjugated FMDV type O or A antibodies (β-glc–O-Ab or β-glc–A-Ab) were treated with the solution containing pan-FMDV antibody-conjugated MNPs (pan-Ab–MNPs) (Scheme 1a) that was reacted with FMDV type O or A, forming a sandwich immunoassay through specific antigen–antibody interaction (Scheme 1b). Finally, res-β-glc was added to the prepared sandwich immunoassay solution for catalytic hydrolysis, and the fluorescent and electrochemical signals were measured by using a fluorescence spectrophotometer and glucometer, respectively (Scheme 1c). The analytical sensitivity, selectivity, and limits of detection (LODs) of the developed biosensor were evaluated, illustrating its ability to perform dual-modal detection.

## 2. Materials and Methods

### 2.1. Materials and Equipment

Res-β-glc, β-glc, bovine serum albumin (BSA), N-hydroxysuccinimide (NHS), tris(2-carboxyethyl)phosphine hydrochloride (TCEP), 2-(N-morpholino)ethanesulfonic acid (MES) buffer, and MNPs were purchased from Sigma-Aldrich (St. Louis, MO, USA). Sulfosuccinimidyl 4-(N-maleimidomethyl)cyclohexane-1-carboxylate (sulfo-SMCC) was obtained from Thermo Fisher Scientific (Waltham, MA, USA). N-(3-Dimethylaminopropyl)-N′-ethylcarbodiimide (EDC) was purchased from Tokyo Chemical Industry (Tokyo, Japan). 1X phosphate-buffered saline (PBS, pH 7.4) solution (Gibco, Thermo Fisher Scientific) was used to dissolve β-glc. Distilled water from a Milli-Q water purification system was used to prepare all chemical solutions. FMDV (pan, O, and A type) antibodies were provided from the Korea Research Institute of Bioscience and Biotechnology. Inactivated antigens of serotype O (O1/Manisa) and A (A22/Iraq) were purchased from The Pirbright Institute (Surrey, UK).

### 2.2. Preparation of pan-FMDV Antibody-Conjugated Magnetic Nanoparticles (pan-Ab–MNPs)

To prepare pan-Ab–MNPs, carboxylated MNPs (7.0 mg) were washed successively with 1 mL deionized water and 1 mL MES buffer (10 mM, pH 6.0), and collected using a magnetic separator. Then, the particles were incubated at 37 °C for 30 min with 1 mL MES buffer (10 mM, pH 6.0) containing a mixture (10 μL) of 800 mM EDC and 1 M NHS, followed by washing three times in the same buffer. Partially carboxylated MNPs were incubated with a 100 μg/mL pan-Ab solution for 1.5 h at 37 °C. Subsequently, the remaining carboxyl-activated groups were blocked by incubation with 1 mL of 1% BSA in MES buffer (10 mM, pH 6.0) for 2 h at 25 °C. Finally, the pan-Ab–MNPs were washed three times with the same buffer and stored at 4 °C for further experiments. The optimal EDC/NHS ratio for efficient antibody conjugation (see Appendix A) was determined by zeta potential measurements (Zetasizer Nano ZS, Malvern Panalytical, Malvern, UK). The conjugation of pan-Ab to the MNP surface was characterized by zeta potential measurements, UV–Vis spectroscopy (Optizen POP, Mecasys, Daejon, Korea), Bradford assays, and in vacuo Fourier transform infrared (FT-IR) spectroscopy (VERTEX 80v, Bruker, Ettlingen, Germany).

### 2.3. Preparation of β-glucosidase-Conjugated FMDV (Type O or A) Antibodies (β-glc–O-Ab and β-glc–A-Ab)

β-glc (0.8 mg) was dissolved in 1 mL 1X PBS solution (pH 7.4) to prepare 6 μM β-glc solution. The solution was incubated at 25 °C for 1 h in 50 μL 1X PBS solution (pH 7.4) containing 1 mM sulfo-SMCC, and then filtered (Amicon^®^ Ultra Centrifugal Filters, 30 K) by centrifugation at 16,000 rpm for 50 min at 4 °C to remove the unreacted sulfo-SMCC. To activate the antibodies, a mixture of FMDV type O or A antibodies (O-Ab and A-Ab, respectively; 0.6 mg/mL) and TCEP (20 μM) was reacted for 2 h at 25 °C to break the disulfide bonds within the antibodies. Finally, to prepare β-glc–O-Ab and β-glc–A-Ab, TCEP-treated O- and A-Ab were treated with β-glc, followed by filtering and incubation for 12 h at 4 °C. The conjugation of β-glc to O- and A-Ab was confirmed by using high-performance liquid chromatography (Ultimate 3000, Thermo Scientific, Waltham, MA, USA).

### 2.4. Optical and Electrochemical Characterization of FMDV

Pan-Ab–MNPs were reacted with various concentrations of FMDV type O and A for 2 h at 25 °C, respectively, followed by washing three times using PBST solution (1X PBS with 0.1% Tween-20) and collected using a magnetic separator.

Next, β-glc–O-Ab and β-glc–A-Ab were successively reacted with pan-Ab–MNPs for 2 h at 25 °C, forming MNP-based antigen–antibody aggregations (Scheme 1b), followed by washing three times using PBST solution and a magnetic separator. Finally, res-β-glc was added to the solution and reacted for 30 min at 37 °C. The fluorescence was measured in a 384 well-plate using a Cytation5 multimode reader (BioTek, Winooski, VT, USA) and the electrochemical signals were measured using a portable glucometer with disposable glucose strips (Accu-Chek Performa, Roche, Basel, Switzerland). UV–Vis spectrophotometry (Optizen POP, Mecasys, Daejon, Korea) and in vacuo Fourier transform infrared (FT-IR) spectroscopy (VERTEX 80v, Bruker, Ettlingen, Germany) were performed to acquire the optical and IR spectra.

## 3. Results and Discussion

### 3.1. Pan-FMDV Antibody-Conjugated Magnetic Nanoparticles (pan-Ab–MNPs)

The conjugation of pan-Ab to the surface of the MNPs was characterized by zeta potential measurements, UV–Vis spectroscopy, Bradford assays, and FT-IR spectroscopy. As shown in Figure 1a, the binding between pan-Ab and MNPs was evaluated as a gradation in surface zeta potential. The zeta potential of carboxylated MNPs was −11.87 ± 0.43, whereas those of the EDC/NHS-activated MNPs and pan-Ab–MNPs were −4.20 ± 0.11 and +4.23 ± 0.10 mV, respectively. Hence, EDC/NHS activation changed the strong negative charge of carboxylated MNPs to a relatively weak negative charge, while conjugation with pan-Ab yielded a shift to a positive charge. This confirms that the antibody conjugation was successful. Figure 1b shows the UV–Vis spectra of the MNPs, pan-Ab, and pan-Ab–MNPs. No absorption band was observed in the MNP spectrum; however, after conjugation with pan-Ab, a broad absorption band originating from the antibodies appeared at 245–290 nm. This shift is attributed to the formation of a complex between the MNPs and antibodies, confirming the successful immobilization of pan-Ab onto the MNPs.

To quantify the loading of covalently immobilized antibodies on the MNPs, we drew a standard calibration curve using Bradford assays, as shown in Appendix A. The Bradford assay is a colorimetric protein assay based on an absorbance shift through Coomassie dye-binding, which enables fast and simple protein quantification. The highest antibody loading of 95.2% occurred with an MNP/antibody ratio of 7:1 (Figure 1c). As shown in Figure 1d, the FT-IR spectrum of pan-Ab–MNPs exhibited an absorption band at 548 cm^−1^ corresponding to Fe–O–Fe, which indicates the characteristic absorption of the MNPs, as well as O–H and C=O stretching vibrations at ~1710 and ~3500 cm^−1^ corresponding to COOH, which indicates the chemical bonding of the MNPs. The N–H peak of the CONH group at 3100–3600 cm^−1^ was observed in the spectra of pan-Ab and pan-Ab–MNPs, further verifying the successful immobilization of pan-Ab onto the MNP surface.

### 3.2. Conjugation of β-glucosidase (β-glc) to FMDV Type O and A Antibodies (O- and A-Ab)

The conjugation of β-glc to O- and A-Ab was confirmed by sodium dodecyl sulphate (SDS)-polyacrylamide gel electrophoresis (PAGE) and Native-PAGE. The results are shown in Figure 2a,b, respectively. The antibodies were treated with TCEP to facilitate the conjugation to β-glc, based on the cysteine residues in the antibodies. The use of TCEP as a thiol-free reducing agent, which is widely applied in the reduction of antibody disulfide bonds, resulted in efficient β-glc–antibody conjugation. In the SDS- and Native-PAGE analyses, the bands of β-glc (Lane 1) were spread widely across the lane, while more intense bands were observed at a molecular weight of 50–75 kD. O- and A-Ab showed slight differences in band position, with a more intense band at a molecular weight of 50–75 kD. The bands above 250 kD in Lanes 4 and 7 verified the conjugation of β-glc to the O- and A-Ab. High-performance liquid chromatography (Figure 2c) further confirmed the successful binding between β-glc and O- or A-Ab by observing the UV–Vis peak of the conjugates in front of that of antibodies.

### 3.3. Optical and Electrochemical Dual-Modal FMDV Detection

The efficiency of optical and electrochemical dual-modal sensing toward FMDV detection was evaluated using the optimum conditions described in Section 3.1 and Section 3.2. The detection mechanism was based on the generation of resorufin and glucose molecules via the catalytic hydrolysis of res-β-glc in the presence of β-glc, whereby the glycosidic bonds cleave to form a terminal non-reducing residue of β-d-glucosides and oligosaccharides (Figure 3). Resorufin exhibits pronounced fluorescent and colorimetric signals due to its high fluorescence quantum yield and long excitation/emission wavelength; hence, it is widely used as a responsive probe for various bioactive species [34]. Additionally, the glucose concentration was measured by a portable glucose sensor using glucose strips, in the same way as blood glucose measurements. As the concentration of used FMDV was not exactly known, RT-qPCR analysis was additionally conducted using FMDV types O and A RNAs extracted from FMDV to determine the exact concentration of FMDV (Appendix A). Briefly, to quantify the concentration of virus samples, qPCR was performed with each serotype of positive control plasmid, which was used in the previous report [1]. Then, standard curves were plotted with Cq values according to the amount of plasmids. The concentrations of each type of viral samples were determined by comparison of the Cq value obtained through RT-qPCR using RNA from the inactivated FMDV with standard curves. This dual-modal sensor system using res-β-glc exhibited excellent selectivity for the detection of FMDV types O and A, indicating that it is a suitable enzymatic probe for the detection of virus specimens.

Figure 4 shows the sensitivity for FMDV type O and A detection by the optical and electrochemical (glucose) measurements. The optical measurement exhibited a linear response depending on FMDV concentration, with an *R*^2^ value of above 0.96 for both FMDV serotypes (Figure 4a,b). Moreover, LODs of log(6.7) and log(5.9) copies/mL were obtained for FMDV type O and A, respectively, revealing excellent selectivity [35]. The glucose concentration was also linearly correlated with the FMDV concentration, with an *R*^2^ value of above 0.98 for both FMDV serotypes (Figure 4c,d). LODs of log(6.9) and log(6.1) copies/mL were obtained for FMDV type O and A, respectively. Despite the slight difference in LOD between the optical and electrochemical measurements, the results prove that the dual detection system can quantify the FMDV concentration by measuring the signals from the released resorufin and glucose, with high selectivity compared to β-glc.

The time-dependent fluorescence and electrochemical changes were monitored according to the concentration of FMDV type O and A (Appendix A). Upon the addition of res-β-glc, the fluorescence intensity increased after 30 min, even with a low concentration of FMDV. In contrast, the measurable time of the electrochemical signals was observed 6 h after addition of res-β-glc, with a slight difference in response time. The difference in response time is considered to be caused by the difference in readout mechanisms of the fluorescent and electrochemical measurement systems.

### 3.4. Selectivity Test

The efficiency and selectivity of optical and electrochemical dual-modal sensing toward FMDV detection were evaluated by screening the fluorescent and electrochemical responses over the cross-reactivity between FMDV type O and A. The cross-reaction was conducted by adding β-glc–O- and β-glc–A-Ab to MNP solutions with FMDV type O and A. As shown in Figure 5a,b, the FMDV-concentration-dependent fluorescence signals showed slight cross-reactivity between the FMDV serotypes at high concentrations. On the other hand, the electrochemical signal showed no cross-reactivity at any measured FMDV concentration (Figure 5c,d). Therefore, the dual-modal sensors prepared from O- and A-Ab could clearly detect FMDV type O and A, respectively. This demonstrates that the dual-modal sensing system is highly suitable for distinguishing FMDV serotypes.

To illustrate the advantage of multi-modal sensors, the background noise was considered, as background noise is a typical problem of fluorescence analyses. As expected, the fluorescence signals appeared to contain background noise at high concentrations of FMDV. Nevertheless, the electrochemical signals rarely contained background noise. Thus, the dual-modal sensing system is highly reliable and selective for the FMDV antigen–antibody reaction.

### 3.5. Comparison with Lateral Flow Assay (LFA)

To evaluate the analytical sensitivity of the dual-modal system, LFA, also known as lateral flow immunochromatographic assay or rapid testing, was employed. Colorimetric LFAs are widely used in medical diagnostics for home testing or laboratory use, particularly for on-site rapid diagnosis of infectious diseases such as FMD. LFA kits were prepared by using as-prepared 40 nm gold nanoparticles as a detection signal probe, and antibodies for detecting FMDV. The sensitivity of the LFAs for FMDV types O and A was compared to that of the dual-modal system based on fluorescent and electrochemical detection. The data was quantified using ImageJ program. As shown in Figure 6 and Appendix A, the LFA sensitivity was estimated to reach to log(6.9) and log(6.1) copies/mL for FMDV type O and A, respectively. Therefore, our dual-modal sensor has a distinctly higher sensitivity than the LFA by both the fluorescent and electrochemical methods in terms of quantitative measurement results.

## 4. Conclusions

In summary, we report a highly effective dual-modal sensing system for FMDV using optical (fluorescent) and electrochemical (glucose) detection methods by using res-β-glc and β-glc. In this approach, FMDV was reacted with pan-Ab functionalized on the surface of MNPs, after which they were treated with β-glc–O- or β-glc–A-Ab. Res-β-glc is catalytically hydrolyzed by β-glc and releases fluorescent resorufin and glucose molecules. The detection efficiency for the two products was evaluated by using a fluorescence spectrophotometer and portable glucometer, respectively. The sensitivity for FMDV types O and A reached log(6.7) and log(5.9) copies/mL through fluorescence measurements, respectively, and log(6.9) and log(6.1) copies/mL through electrochemical measurements, respectively. Importantly, we confirmed that O- and A-Ab-sensitized systems could clearly distinguish between FMDV types O and A in the dual-modal system with high selectivity. Thus, the proposed dual-modal sensing system offers improved sensitivity and selectivity for FMDV detection. ELISA, RT-PCR, and RT-LAMP, widely used to diagnose FMDV, often suffer from false-positive reactions. However, our dual-modal sensing system has the advantage of being able to reduce such fatal errors generated from single diagnosis methods. Even though, compared to fluorescent detection, it was confirmed that the electrochemical signal could not be measured to a lower concentration due to the limitation of the performance range of the glucometer, our strategy has the possibility to be improved with sensitive detecting devices. This study provides an easy assay for the highly reliable detection of FMDV in real samples. In addition, it presents a strong example of the potential of dual-modal sensing for the development of portable point-of-care devices and next-generation diagnostic platforms.

## Data Availability

Not applicable.

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
