# Peer review of "Effective Diagnosis of Foot-And-Mouth Disease Virus (FMDV) Serotypes O and A Based on Optical and Electrochemical Dual-Modal Detection"

_biomolecules, 2021, doi:10.3390/biom11060841_

Round 1

Reviewer 1 Report

I reviewed the manuscript entitled “Optical and electrochemical dual-modal detection of foot-and mouth disease virus (FMDV). In this study the authors show the development of a diagnostic method for FMDV based on the use of optical (fluorescent) and electrochemical (glucose) detection methods.

Overall, based on the tremendous economic impact that FMDV has in the livestock production around the world, I consider as relevant the results presented in this study. However, I consider that some issues in the manuscript should be properly addresses to improve its quality for publication.

  1. Considering the potential use that this technology may have in the diagnostic of FMDV, I suggest the authors to modify the titer of its study for something like: “Rapid diagnostic of foot-and mouth disease virus (FMDV) serotypes A and O based on Optical and electrochemical dual-modal detection.
  2. Between lines 46 and 47 when you are describing FMDV as a virus to produce a fast infection rate and high mortality rate, I suggest changing for: high morbidity and low mortality rates. Mortality in FMDV is sometimes associated mostly with infections in young animals where the myocarditis is developed during the clinical course of the disease. Also, in this section between lines 57 and 66, authors describe different diagnostic methods for FMDV. In my opinion it should be highlighted that the proper diagnosis of FMDV is based on the combination of multiple diagnostic tests. There is not a perfect test, and considering the economic impact of FMD, the confirmation of the presence of FMDV should be carried out based on that strategy. In this context, despite the limitations described by the authors for the use of PCR techniques, there are valuable tools for the diagnosis of FMDV. I suggest the authors to add a reference to support the statement about the frequently generation of false positive results by PCR techniques in FMDV. Also, I consider that a reference should be added to support statement between lines 54 and 56.
  3. I suggest the authors to describe more details about the viruses used in this study like strain source of isolation etc. Also, I suggest including more information about the PCR methodology used to quantify the copy number of these viruses.
  4. Although methods and discussion sections were merged, I consider that more details about this study should be discussed. I suggest including more information like applicability in the field. For FMD different kind of samples may be use for the diagnosis of FMDV like epithelial samples, probang samples, viremic blood samples or viral isolates, the method presented for the authors could be use with all kinds of samples. Can this test be used as a rapid test in the field? Or it must be performed in the lab? Do authors see any limitation in affected livestock species like sheep, a specie that developed less epithelial lesions during the infection? Mention some advantages and disadvantages of the use of this technology for the diagnostic of FMDV, and how it can complement the current methodology used to diagnose FMDV.

Author Response

Review Report

We are thankful for valuable comments of the reviewers, which helped us to correct errors and strengthen up our manuscript.

Reviewer #1 ;

I reviewed the manuscript entitled “Optical and electrochemical dual-modal detection of foot-and mouth disease virus (FMDV). In this study the authors show the development of a diagnostic method for FMDV based on the use of optical (fluorescent) and electrochemical (glucose) detection methods.

Overall, based on the tremendous economic impact that FMDV has in the livestock production around the world, I consider as relevant the results presented in this study. However, I consider that some issues in the manuscript should be properly addresses to improve its quality for publication.

  1. Considering the potential use that this technology may have in the diagnostic of FMDV, I suggest the authors to modify the titer of its study for something like: “Rapid diagnostic of foot-and mouth disease virus (FMDV) serotypes A and O based on Optical and electrochemical dual-modal detection.

Response : As the reviewer suggested, we modified the title to “Effective diagnosis of foot-and mouth disease virus (FMDV) serotypes A and O based on Optical and electrochemical dual-modal detection.”, because this work is focused on ‘effective diagnosis’ rather than ‘rapid diagnosis’.

  1. Between lines 46 and 47 when you are describing FMDV as a virus to produce a fast infection rate and high mortality rate, I suggest changing for: high morbidity and low mortality rates. Mortality in FMDV is sometimes associated mostly with infections in young animals where the myocarditis is developed during the clinical course of the disease. Also, in this section between lines 57 and 66, authors describe different diagnostic methods for FMDV. In my opinion it should be highlighted that the proper diagnosis of FMDV is based on the combination of multiple diagnostic tests. There is not a perfect test, and considering the economic impact of FMD, the confirmation of the presence of FMDV should be carried out based on that strategy. In this context, despite the limitations described by the authors for the use of PCR techniques, there are valuable tools for the diagnosis of FMDV. I suggest the authors to add a reference to support the statement about the frequently generation of false positive results by PCR techniques in FMDV. Also, I consider that a reference should be added to support statement between lines 54 and 56.

Response : As the reviewer suggested, we modified ‘a fast infection rate and high mortality rate’ to ‘high morbidity and low mortality rates’(lines 48). Also, we added the references for sentences of line 55-56 (reference #4) and lines 64-66 (reference #18,19) to support statement about the recent outbreak of FMD in South Korea and frequently generation of false positive results by PCR techniques in FMDV, respectively.

  1. I suggest the authors to describe more details about the viruses used in this study like strain source of isolation etc. Also, I suggest including more information about the PCR methodology used to quantify the copy number of these viruses.

Response : As the reviewer commented, we added reagent type of FMDV (line 130-132).

“Inactivated antigens of serotype O (O1/Manisa) and A (A22/Iraq) were purchased from The Pirbright Institute (Surrey, UK).”

In addition, we added more information about the PCR methodology to quantify the copy number of viruses in the text in the revised manuscript (line 241-246).

“Briefly, to quantify the concentration of virus samples, qPCR was performed with each serotype of positive control plasmid which was used in the previous report [1]. Then, standard curves were plotted with Cq values according to the amount of plasmids. The concentrations of each type of viral samples were determined by comparison of the Cq value obtained through RT-qPCR using RNA from the inactivated FMDV with standard curves.”

  1. Although methods and discussion sections were merged, I consider that more details about this study should be discussed. I suggest including more information like applicability in the field. For FMD different kind of samples may be use for the diagnosis of FMDV like epithelial samples, probang samples, viremic blood samples or viral isolates, the method presented for the authors could be use with all kinds of samples. Can this test be used as a rapid test in the field? Or it must be performed in the lab? Do authors see any limitation in affected livestock species like sheep, a specie that developed less epithelial lesions during the infection? Mention some advantages and disadvantages of the use of this technology for the diagnostic of FMDV, and how it can complement the current methodology used to diagnose FMDV.

Response : As the reviewer suggested, we added description regarding discussion on the advantages of our diagnostic system compared to the methods currently used for FMDV in the revised manuscript (lines 262-268).

“ELISA, RT-PCR and RT-LAMP, widely used to diagnose FMDV, often suffer from false-positive reactions. However, our dual-modal sensing system has the advantage of being able to reduce such fatal error generated from single diagnosis methods. Even though, compared to fluorescent detection, it was confirmed that the electrochemical signal could not be measured to a lower concentration due to the limitation of the perfor-mance range of the glucometer, our strategy has the possibility to be improved with sensi-tive detecting devices.”

Reviewer #2 ;

The authors presented a dual-mode selective and sensitive detection for Foot-and-mouth disease virus (FMDV). The paper is overall very interesting and clear and is recommended for publication after addressing the following major concerns:

  1. Is a capsid protein used as a target molecule? If so, how the authors have expressed the LODs in viral copies/mL instead of using the molarity of the analyte? How many capsid proteins covers one virus?

Response : Actually, the anti-FMDV antibodies used in this study bound to the capsid proteins. However, it is known that one FMDV particle is composed of icosahedral capsid consisting of four structural proteins, VP1, VP2, VP3 and VP4. Also, commercial FMDV particles used in this study may not exist as fully intact form, because it goes through the process of inactivation. Therefore, we had a difficulty in the determination of concentration of FMDV particles with no information provided from The Pirbright Institute. Thus, to quantify the concentration of FMDV particles, we performed qRT-PCR and measured the concentration of viral samples by comparison of Cq value with positive controls. In this way, we expressed the LODs in viral copies/mL, not molarity of capsid proteins. More details were described in the revised manuscript (line 241-246).

“Briefly, to quantify the concentration of virus samples, qPCR was performed with each serotype of positive control plasmid which was used in the previous report [1]. Then, standard curves were plotted with Cq values according to the amount of plasmids. The concentrations of each type of viral samples were determined by comparison of the Cq value obtained through RT-qPCR using RNA from the inactivated FMDV with standard curves.”

  1. How the LOD have been evaluated? could the authors briefly describe it?

Response : As shown in Figure 4, we evaluated the sensitivity through the optical and electrochemical measurements for each type of FMDV (O and A). Based on these results, we calculated the LOD through ‘Blank + 3SD’. We added a reference (reference #36, page 7) to support statement about how we obtain the LOD.

“Moreover, LODs of log(6.7) and log(5.9) copies/mL were obtained for FMDV type O and A, respectively, revealing excellent selectivity [36].”

Reviewer 2 Report

The authors presented a dual-mode selective and sensitive detection for Foot-and-mouth disease virus (FMDV). The paper is overall very interesting and clear and is recommended for publication after addressing the following major concerns: 

1) Is a capsid protein used as a target molecule? If so, how the authors have expressed the LODs in viral copies/mL instead of using the molarity of the analyte? How many capsid proteins covers one virus?

2) How the LOD have been evaluated? could the authors briefly describe it?

Author Response

(The authors gave the same response as above.)

Round 2

Reviewer 2 Report

The authors have addressed all the reviewer comments